# Discrete Fourier Transform with Denoise Model Based Least Square Wiener Channel Estimator for Channel Estimation in MIMO-OFDM

**DOI:** 10.3390/e24111601

**Published:** 2022-11-03

**Authors:** Dhanasekaran S, SatheeshKumar Palanisamy, Fahima Hajjej, Osamah Ibrahim Khalaf, Ghaida Muttashar Abdulsahib, Ramalingam S

**Affiliations:** 1Department of E.C.E., Sri Eshwar College of Engineering, Coimbatore 641202, India; 2Department of E.C.E., Coimbatore Institute of Technology, Coimbatore 641014, India; 3Department of Information Systems, College of Computer and Information Sciences, Princess Nourah bint Abdulrahman University, P.O. Box 84428, Riyadh 11671, Saudi Arabia; 4Al-Nahrain Nanorenewable Energy Research Center, Al-Nahrain University, Baghdad 64074, Iraq; 5Department of Computer Engineering, University of Technology, Baghdad 10066, Iraq

**Keywords:** MIMO, OFDM, least square estimator, channel estimation, minimum mean square error, discrete Fourier transform

## Abstract

Multiple-input Multiple-Output (MIMO) systems require orthogonal frequency division multiplexing to operate efficiently in multipath communication (OFDM). Channel estimation (C.E.) is used in channel conditions where time-varying features are required. The existing channel estimation techniques are highly complicated. A channel estimation algorithm is needed to estimate the received signal’s correctness. In order to resolve this complexity in C.E. methodologies, this paper developed an Improved Channel Estimation Algorithm integrated with DFT-LS-WIENER (ICEA-DA). The Least Square (L.S.) and Minimum Mean Square Error (MMSE) algorithms also use the Discrete Fourier Transform (DFT)-based channel estimation method. The DFT-LS-WIENER channel estimation approach is recommended for better BER performance. The input signal is modulated in the transmitter module using the Quadrature Phase Shift Keying (QPSK) technique, pulse modeling, and least squares concepts. The L.S. Estimation technique needs the channel consistent throughout the estimation period. DFT joined with L.S. gives higher estimation precision and limits M.S.E. and BER. Experimental analysis of the proposed state-of-the-art method shows that DFT-LS-WIENER provides superior performance in terms of symbol error rate (S.E.R.), bit error rate (BER), channel capacity (CC), and peak signal-to-noise (PSNR). At 15 dB SNR, the proposed DFT-LS-WIENER techniques reduce the BER of 48.19%, 38.19%, 14.8%, and 14.03% compared to L.S., LS-DFT, MMSE, and MMSE-DFT. Compared to the conventional algorithm, the proposed DFT-LS-WIENER outperform them.

## 1. Introduction

Enabling 5G technology is recognized by MIMO in mobile communications [1]. Performance gain in extensive MIMO can be attained, and reliable exposure to channel state information (C.S.I.) is essential [2]. A large number of downlinks to C.S.I. are needed to acquire C.S.I. conventionally, since they can use the large-scale array antenna at the base station, even for a massive downlink MIMO [3,4,5,6,7]; then, since the array size of the base station is proportional, the channel matrices are not able to compute and this requires extreme maximized training for downlink overheads since it is a direct estimation [8].

MIMO transmission will be a significant part of wireless communication in the coming days in wireless technology. This type of system has minimum difficulty executing OFDM and is simple. In contrast, in the case of non-orthogonal multi-carriers, the MIMO structure of the receiver has become more complicated since it has to handle various interference measurements, i.e., inter-symbol interference (I.S.I.), inter-carrier interference (I.C.I.), and inter-antenna interference (I.A.I.) [9,10,11]. Those reference signals that are received have been influenced by various measurements of interference (known as pilots); this estimates the channel (C.E.) and the complicated and consecutive equalization (E.Q.) of the received signal (Rx). Mostly, the channel is dispersed in multiple types of broadband communication [12,13,14,15].

In ref. [4], it was found that the improved channel estimation of C.P. information could be designed to be utilized through Kalman filters. However, this technique can be employed only for orthogonal waveforms. In order to improve the performance of Channel Estimation by reusing pilots’ knowledge from C.P., this paper utilized the localization of symbol-time in orthogonal waveforms [16,17].

It can also be referred to by correctly localizing the pilot energy up to the terminated point of the block. It needs to possess maximum energy while it is copied to the C.P. Identification of symbols in the data of non-orthogonal multi-carriers and acquires the various Channel Estimation and C.Q. modules independent of the usual practice. The earlier works on Channel Estimation [18] were calculated with L.S. then LMMSE estimators that assumed that I.S.I. and I.C.I. manage additional correlation noise with the white noise of non-orthogonal waveforms. Even though the non-iterative techniques have been discussed in [19], they can acquire the implementation of Channel Estimation with increased errors and poor SNR. It is directly applicable to MIMO interference with little interference and non-orthogonal waveforms and has no solution for channel estimation.

Channel Estimation (C.E.) utilizes C.S.I. to transmit and receive the wireless signal. In a wireless communication medium, C.S.I. causes power degradation, fading, and wave scattering as a function of transmission distance. The channel state information (C.S.I.) is determined by comparing the pilot signals used during the transmitter to the received pilot signals. In contrast, the SNR reflects the resilience of the bit stream about the channel noise. The concept of C.E. is evolving based on the objective of the correlation of signal at the center point integrated with suppression of the M.A.I. This signal quality impacts receiver BER performance [20]. In C.E., two schemes are employed for estimating signals, namely, (i) pilot-assisted scheme, which trains the allocation of symbols to bandwidth; and (ii) blinding using statistical features. At the receiving end, channel estimation inserts pilot signals into the transmitted signal, resulting in signal overhead, which a minimum number of pilot symbols can reduce.

Currently, for uplink MU-MIMO communication, countless MIMO detectors are used. In addition, an optimal maximum likelihood detector and a low-complexity environment are also developed.

The channel estimate is crucial in OFDM systems. Enhancing the system’s bit error rate performance is utilized to boost the channel capacity of an OFDM system. Based on a comb-shaped pilot array, channel estimation is performed on the pilots, and a Rayleigh fading channel is considered.

A performance measure of the M.S.E. of the channel estimator fusing DFT-based procedure obtains the best results as contrasted with the consolidation of LS-based strategies. The comparative analysis is carried out by varying parameters, such as the number of subcarriers, channel taps, cyclic prefix length, and pilot frequency.

This article recommends a channel estimation method incorporating differential evolution and L.S. with DFT. The proposed ICHE-DA is included in the central part of the OFDM-MIMO block. We consider a two-user interference channel for MIMO-OFDM systems. We demonstrate the basic idea of a user-based transmitter and receiver using DFT-LS-WIENER. Further, we focus on interference from frequency-selective channels. By doing so, the main struggle is to evaluate the accuracy of covariance matrices, which may need accurate information on interference parameters. Moreover, one of the contributions is to analyze the firmness of the spectrum by coinciding with the DFT series. The findings showed that the suggested system outperforms the current technique in terms of BER, PSNR, CC, and MMSE.

### Problem Statement

In an ideal radio channel, there would only be one direct path signal, a flawless recreation of the transmitted signal. On a natural channel, the signal is altered during transmission. Copies of the transmitted signal that have been attenuated, reflected, refracted, and diffracted make up the received signal. Additionally, the channel puts noise into the signal and causes the carrier frequency to shift as the transmitter or receiver moves (Doppler effect). A radio system’s performance depends on the radio channel’s characteristics. Therefore, understanding these effects on the signal is crucial. As a result, in our work, we consider a MIMO-OFDM system before employing pilot permutation to estimate the channel. This article compares the L.S., MMSE without DFT, and the L.S., MMSE with DFT. A new DFT-LS-WIENER hybrid channel estimation technique is also proposed to improve bit error rate performance.

The rest of the paper is organized as follows: Section 2 describes the literature survey. Section 3 presents the proposed DFT-LS-WIENER method. Section 4 discusses the comparison of the proposed work. Finally, Section 5 presents the conclusion

## 2. Literature Survey

In their introduction of the MIMO-OFDM for hybrid channel estimator, Nie et al. [3] underlined the sparse nature of angular channels and the use of compressed sensing (C.S.) tools based on wave systems. The continuous annular channel has resolved the issue of restricted angular resolution. The discretization has also been resolved by the angular support of sparse channel designs utilized in C.S. based on channel estimation. The proposed channel estimator’s minimum construction error and optimal performance are calculated by performance analysis [20]. Lee et al. [7] discussed a novel scheme for channel estimation in downlink channels using the CSMBCS technique in the downlink channel estimation. The usual sparsity and cluster design have been employed where the user is unknown to evaluate the scenario of various subcarriers of downlink channels. The initial local clusters have been constructed using the Bayesian architecture described by sparsity and local beta process (LBP) property. However, accuracy may be low [21].

Qiao et al. [8] determined the parallel-interference-cancellation (P.I.C.) technique based on MIMO LMMSE for shared channel estimation and leveling non-symmetrical waveforms. In contrast to the essential practice, appropriately restricting the pilots in time, and area, likewise utilizing pilots’ data from C.P. Using execution findings showed that using C.P. data from pilots resulted in edge error rate execution increases of up to 2.4 dB above the OFDM signal. It performs poorly in BER tests [22]. Berraki et al. [10] proposed a semi-daze, time–space estimation of channel procedure for frequency-selective enormous MIMO frameworks. The arrangement relies upon subspace spread by signing eigenvectors obtained from a signing covariance matrix. Critically, the receiver examines the reception rate based on time–space estimation accessed for an exact matrix estimation. The estimation technique does not need symmetry between preparation images of clients in all cells [23].

Alkhateeb et al. [11] anticipated the NAMP calculation as an assessment model that depends on the LTE-Advanced remote channel model. At first, NAMP need not bother with the sparsity level prior information. At that point, the fixed advance size that improves the sign reproduction effectiveness is decided. Lastly, a Singular Entropy request assurance component endeavors to avoid the less pertinent iotas that can be presented. It accomplishes increasingly stable execution, particularly in low SNR channel conditions, and after that, the multifaceted computational nature is decreased, such as the versatile SAMP calculation. The main drawback of this work is that it has higher computational complexity [24].

The estimation of the channel covariance matrix of the received sampled signals in the frequency domain is proposed by Ngo et al. [25]. An asymptotically massive MIMO system’s channel vector possesses orthogonal property, where the channel vector and eigenvector are proportional. The covariance matrix estimation by a limited number of signal samples produces an error and the assumption that channels are perfectly orthogonal are the two limitations of the system. The condition assumed in this method is that the numbers of both the samples and antennas are asymptotically large.

Zarifi et al. [26] proceeded with a CDMA system having unknown WSS noise. The centro-Hermitian property estimates the generalized correlation decomposition-based blind channel estimation noise covariance matrix for estimating noise subspace in the data samples. Lama et al. [27] considered multipath channels for the STBC-CDMA system. STBC is implemented by exploiting spreading codes with unique structures and developing a computationally efficient channel estimation algorithm.

The angular resolution is improved by utilizing redundancy samples containing substantially more uniform DSFT premises than the antenna numbers proposed by Lee et al. [28]. DSFT number premises are equal to angular spectrum samples at an increased rate, which increases both lengths of the short channel vector that need to be recuperated, and the number of nonzero components improved. Along these lines, more channel estimations (longer preparing groupings) should be acquired for extraordinary recuperation of the signal.

Li et al. propose the Minimum Mean Square Error (MMSE) scheme with second-order statistics [29]. The limitations of this method are the channel estimation error. A joint estimation of channel length and OFDM system impulse response is proposed by Charles et al. [30], using the balanced achievement between information, specification based on Kullback–Leibler divergence, and noise rejection with an accurate channel description. Hence, data specification is unsuitable for practical channel length estimation techniques because of prohibitive complexity. Estimating the channel is performed recursively, allowing optimal channel length establishment with considerable cost and increased performance and robustness. The main limitation of the proposed work is that power consumption is high. Dhanasekaran et al. [31] presented a game theory-based MIMO-OFDM communication system for reducing the BER and NMSE value. This work improves the performance of the MIMO-OFDM system [32,33,34,35,36]. Ramalingam et al. introduced various communication protocols using optimization techniques and IoT networks [31,37,38,39,40,41,42].

## 3. Proposed Methodology

This work presented an improved channel estimation algorithm DFT-LS-WIENER integrated with a differential approach. Moreover, this paper uses the L.M.DFT approach for channel estimation in the wireless communication medium. The middle area of the MIMO framework is where the suggested DFT-LS-WIENER is located. MIMO systems typically include several transmitters and a receiver antenna for parallel data transmission. The proposed model is examined for two transmitters and four-receiver antennae. Quadrature Phase-Shift Keying modulation (QPSK) is utilized for input modulation on the transmitter side [43,44,45,46,47]. Then, I.S.I. is mitigated through the utilization of P.S.A., where P.S.A. is involved in transmitting waveform pulses and effectively utilizing available bandwidth. 

If signal bandwidth is more effective than channel bandwidth, it leads to signal distortion. This distortion is observed in the I.S.I. environment. To control I.S.I. in wireless communication, a pulse shaping filter (P.S.F.) is utilized for analysis. On each transmitter side, symbol mapping and IFFT operations are performed. Subsequently, this leads to transmitting information over the Multipath channel and includes AWGN noise on the receiver side. At the receiver, this operation is performed inversely with the utilization of DFT-LS-WIENER. With the utilization of LMMSE and DFT, a semi-blind review is performed. Figure 1 details the overall flow of constructing a MIMO system with ICEA-DA.

### 3.1. System Description

All of the subcarriers have been modulated using different techniques. The results have been simulated using the QPSK modulation technique. The next step in aligning the bit stream over the subcarriers is the mapping of the subcarriers. In the OFDM design, the Wiener filter is employed because it reduces the mean square error between the actual transmitted signal and the received signal. Here, L = 7 is the Wiener filter order under consideration. Initially, the incoming signal information is transmitted through pilot sequences and modulation schemes such as the QPSK scheme [48,49]. The QPSK scheme consists of the advantages and noise factor. Another advantage of this scheme is a higher information transmission rate than another modulation scheme with minimal bandwidth.

The signal shift consists of ‘4’ signal states with the transmission of information through the QPSK scheme as 2 bits/symbol. Each signal in the system is modified by 1 bit thanks to the encoding strategy. The transmitted signal is represented by the sine and cosine of the following Equation (1):(1)$$g_{n}\left(t\right)=\sqrt{\frac{2S_{e}}{D_{s}}}\cos\left(2\pi f_{c}t+\left(2n-1\right)\frac{\pi }{4}\right),n=1,2,3,4$$

Here, $g_{n}\left(t\right)$ signifies the signal based on the time, $S_{e}$ represents the symbol energy, symbol duration is denoted as $D_{s}$, whereas $f_{c}$ is defined as the baseband of the signal.

In the modulator of the MIMO scheme, 4 phases of the signal are generated, which consists of 2D signal space with unit function $\varphi_{1}\left(t\right)$ and $\varphi_{2}\left(t\right)$ denoted as follows Equations (2) and (3)
(2)$$\varphi_{1}\left(t\right)=\sqrt{\frac{2}{D_{s}}}\cos2\pi f_{c}t,\;0\leq t\leq T$$
(3)$$\varphi_{2}\left(t\right)=\sqrt{\frac{2}{D_{s}}}\sin2\pi f_{c}t,\;0\leq t\leq T$$

From the above, Equations (2) and (3) $\varphi_{1}\left(t\right)$ and $\varphi_{2}\left(t\right)$ are used to measure the quadrature and in-phase components. According to Equation (4), an OFDM signal is conceptually composed of four signal constellations based on four points in the signal space.
(4)$$\left(\pm \sqrt{\frac{S_{e}}{2}},\pm \sqrt{\frac{S_{e}}{2}}\right)$$

Here, the system’s full power, distributed equally between the two carriers involved, is represented by the signal factor 1/2. At the receiving end, the symbols are demodulated by removing the carrier phase factor, and the continuously received robot in phase 2 is determined from the input data.

Let $p_{t}\left(t\right)$ indicates the transmit-side P.S.F. for each symbol and $p_{r}\left(t\right)$ signifies the receive antenna side matched the filter. The composite channel is symbolized as H(t). The $\left(I_{R},I_{T}\right)$ time-domain signal received at the $i_{R}$ antenna is as follows:(5)$$h_{iR,iT}\left(n\right)=\sum_{i_{T}=1}^{N_{T}}h_{iR,iT}\left(n\right)⊗x_{iT}+N$$

Here, Equation (5) represents/denotes noise placed in the uncorrelated spatial and temporal region and/or represents the presence of noise in transmitting the time-domain signal at their antenna. After the removal of I.S.I., with the utilization of IFFT, ISI is eliminated. Moreover, it uses the symbol mapping technique at each transmitter side. Fourier analysis is performed in frequency domain components for signal transmutes in time or spatial domain factors. IFFT is implemented at the receiver end to improve flexibility, execution speed, and precision. Different complexes and data points are transformed for the same number of time-domain points when the IFFT procedure is used. The IFFT involved in the execution of N-point receiver operation in Equation (6) IFFT is presented mathematically as follows: (6)$$u\left(n\right)=\frac{1}{N}\sum_{n=1}^{N-1}U\left(k\right)e^{\frac{-j\pi kn}{N}},n=0,1,.....,N-1$$

The represented data frame transforms where/the size or the number of signal points. *k* = 0, 1, …, *N* − 1, and U(k) indicates the FFT frequency output at the *k*th end. Then, FFT analysis transforms the signal in the receiver block in the time domain. The following is the mathematical formulation for the FFT in Equation (7).
(7)$$U\left(k\right)=\sum_{n=1}^{N-1}u\left(n\right)e^{\frac{-j2\pi kn}{N}},k=0,1,.....,N-1$$

Moreover, the information is passed in a multipath channel environment with added AWGN noise characteristics. Instead of the message signal, AWGN is used in this document due to its higher bandwidth. The example that follows shows how system n’s noise generates a vector that has the same components as the message defined in Equation (8) and a random stationary point with mean 0 of a complex Gaussian distribution:(8)$$m=n_{p}\times \left(M\left(0,1\right)\right)+i\times M\left(0,1\right)$$
where $n_{p}$ signifies the noise power and M(0,1) represents a message signal of the same length. It includes random variables that are normal or Gaussian.

### 3.2. Firmness of Spectrum

When a signal is delivered, the fundamental frequency is f_0_ = 1/T0, where T0 is the signal’s period. This translates to a frequency of 1/N for discrete-time transmissions. Since all other frequencies are multiples of this fundamental frequency, the resolution of the spectrum is determined by the number of samples N. At *k* = N/2, it is possible to determine the highest signal frequency that X[k] may represent. According to Shannon’s sampling theorem, the highest frequency of the original signal, which is assumed to be an audio stream sampled at 44.1 kHz, is 44.1/2, or 22.05 kHz. In the case of N = 1024 samples, the coefficient at *k* = N/2 corresponds to a frequency of 22.05 kHz, and the coefficient at *k* = 1 to a frequency of 22.05/512 43.1 Hz. The DFT coefficients would represent various frequencies (for instance, *k* = 1 corresponds to 22.05/500 = 44.1 Hz) if we sampled another area of an audio signal with just 1000 samples (at the same sample rate). According to the conclusion, the spectra can only be compared if the DFT spectra are computed using the same number of samples. The second numerical sample demonstrated another impact on the resolution. The DFT spectrum cannot fail to reflect the genuine contribution of the individual frequencies of an arbitrary signal x[n], even if the signal is periodic because it is sampled, i.e., it only has information about particular discrete frequencies.

### 3.3. Receiver Structure

A receiver with channel estimation and MIMO detection is proposed using LS-DFT with overlapping pilots. Without sacrificing generality, each subcarrier in each frame-block is assumed to have a pilot, i.e., NF = NT = 1, yielding N(T.S.) = N.N.T., and pilot existence of multiple or redundancy N.R. = N (T.S.)/N.G. The phase difference of the received signal causes multipath fading. This is because the signal strength moves over varying distances and paths. Use a Rayleigh distribution to reach the receiver via multiple paths. For fast-fading situations, the Rayleigh distribution is most commonly used. The pilots are assigned by reusing the pilot sequence from the multiple request pilot requests themselves. The multi-objective optimization algorithm can easily optimize the time-varying channel metric. The proposed pilot allocation method has the potential to provide a powerful method for counting the sends with a request to users within their cells. The fitness function (F.F.) is a crucial component of evolutionary algorithms. The pilot pattern of multiple pilot requests is reused in this document for the pilot assignment. The optimization algorithm can optimize the channel measure that changes over time.

The proposed pilot allocation method has the potential to provide a powerful method for counting the requesting users in their cells. The fitness function (F.F.) is critical to the receiver-side optimization concept.

In order to develop a solution for parallel search optimization and the continuous process, the DFT-LS-WIENER technique is proposed. Because control parameters are included, the suggested DFT-LS-WIENER offers improved convergence benefits. With efficient continuous variable handling and integer variable optimization, the suggested DFT-LS-WIENER is provided in canonical form.

### 3.4. Least Square Error (L.S.) Estimation

If ‘Z’ is sent through a channel ‘j’ such that it can be written in matrix form as,
y = Z_j_(9)

Then, the error ‘e’ can be defined as
e = x″ − x(10)
where x is the anticipated result.

The definition of the squared error (S) is
S = |e|2 S = (x″ − x)2 S = (x″ − x) ∗ (x″ − x)t(11)
where “t” in superscript denotes complex matrix transposition.
S = (x″ − Zj) ∗ (x′′ − Zj)t(12)

By taking the derivative with respect to ‘j’ and setting it equal to zero, we may minimize this equation. The result of the final equation is:j″ = (Zt Z) − 1Ztx(13)
j″ = Z − 1 x j ls = Z − 1x(14)

Both SISO and MIMO systems can use Equation (14).

### 3.5. DFT-LS-Wiener Channel Estimation

At the pilot frequencies, channel estimation is performed using the Least Squares (L.S.) approach. In this case, DFT-based channel estimation is used. As a result, the channel answers are shortened to the number of channel taps before being approximated using the least square method.

After completing the DFT-based LS estimation, the estimates are fed to the Wiener filter model.The goal of applying the Wiener filter model in channel estimation is to decrease the quantity of noise in a noise-affected signal and to decrease the mean square error between the actual broadcast signal and the received signal.Wiener coefficients are produced based on the order of the Wiener filter; for example, if the order is six, then the Wiener filter model produces six Wiener coefficients.DFT-LS estimates are filtered using these Wiener coefficients.The desired output is a noiseless channel estimate.

By applying linear time-invariant (L.T.I.) approaches to filter a noisy experimental process, a Wiener filter is a signal processing filter that generates an estimate of a desired or target stochastic process. The Wiener filter decreases the mean squared error between the estimated and predicted random processes. By using the pertinent signal as input and filtering the known signal to obtain an estimate as output, a Wiener filter aims to compute a statistical estimate of an unknown signal. For instance, an interesting unknown signal tampered with by additive noise can be present alongside a known signal. It is possible to estimate potentially important signals using Wiener filters to remove noise from distorted signals. A typical deterministic filter’s frequency response is designed for the selected carrier frequency. The Wiener filter, however, is created differently. To develop a linear time-invariant filter whose output is as close as possible to the original signal, one must be aware of the spectral properties of both the original signal and noise.

### 3.6. Finite Impulse Response Wiener Filter Model for Discrete Series Ls Estimates

The output of the Wiener filter g[n] is compared with the reference signal s[n] after being convolved with the input signal H_l_s[n]. The DFT-LS estimates (H_l_s[n]) are supplied to a Wiener filter with coefficients a_0_, a_1_…a_N_ of order (number of touches passed) N in order to produce the Wiener filter coefficients. The expression’s result, x[n], is used to signify the filter’s output. The difference between the estimated and original signals is reduced to a minimum via the Wiener filter model. It significantly reduces root mean square error and improves bit error rate performance. The block diagram of the F.I.R. Wiener filter with DFT_LS estimates as input is shown in Figure 2.

## 4. Results and Discussions

This research was conducted to determine and compare the proposed algorithm’s performance to that of other systems. To evaluate the proposed approach system performance, the measures that will be evaluated are BER, M.S.E., and MMSE. The proposed algorithm measures network performance by varying the system’s SNR concerning the number of receiver and transmitter antennas. The suggested algorithm is run using machine configuration on the MATLAB platform.

### 4.1. Performance Analysis

Table 1 shows the system configuration used in the developed model and its values.

Here, the proposed ICEA-DA with LS FFT is utilized to analyze the system’s performance. The metrics considered for analysis are expressed as follows:

#### 4.1.1. Bit Error Rate (BER)

In characterizing data channel performance, it is used as an essential parameter. At the remote end, errors in critical parameters appear while transmitting data from one to another point in wireless link/radio or wired telecommunication link.
(15)$$BER=\frac{N_{b}\left(e\right)}{T_{N}\left(t_{b}\right)}$$
where $N_{b}\left(e\right)$ is the number of bits in error, and $T_{N}\left(t_{b}\right)$ is the total number of transmitted bits.

#### 4.1.2. Mean Square Error

It is calculated as the average squared deviations between estimated and actual values. Its risk function is equivalent to the predicted values of error function loss.

A comparison of the simulation and analytical results, which yield the M.S.E. and BER of equalization generated by DFT-based then Wiener channel estimators, is shown in Table 2 and Table 3 and Figure 3. The output of the simulation’s MSE-related equalizer is derived using analytical and semi-analytical methods. A minor disagreement was found with high SNR (>15 dB) intended for a DFT-based channel estimator.

Table 4, Table 5, Table 6, Table 7 and Table 8 and Figure 4 and Figure 5 illustrate the L.S. and MMSE channel estimation methods performance of M.S.E. and BER. The experimental results represent the estimator gains of MMSE, which are more than 4 dB larger than the L.S. estimator through all values of SNR, because of the MMSE ability to remove the noise outside. Additionally, if the value of SNR is too high, then the MMSE estimator of M.S.E. becomes low compared to the L.S. estimation.

### 4.2. Simulation Results with Number of Subcarriers Varied

Figure 6 shows that when the number of subcarriers is 64, the bit error rate for the DFT-LS-WIENER technique at 15 dB SNR is 0.02285, which is significantly less than the L.S., LS-DFT, MMSE, MMSE-DFT techniques.

### 4.3. Simulation Results with Length of Cyclic Prefix Varied

Table 9 and Table 10 discuss the BER and SNR performance analysis compared with conventional algorithms at different subcarriers K. The proposed approach provided the minimum BER value at different subcarriers compared to the conventional algorithm. The comparison of the proposed DFT_LS_WIENER approach and the conventional algorithm is shown in Figure 7, Figure 8, Figure 9, Figure 10, Figure 11, Figure 12 and Figure 13.

Figure 10 shows that when the cyclic prefix length is increased to 1/4th of the number of subcarriers, the bit error rate is reduced compared to the previous cyclic prefix length (1/8th of the number of subcarriers).

### 4.4. Simulation Results with Pilot Frequency Varied

Table 11 shows the BER of SNR with pilot frequency varied.

### 4.5. Analysis

The performance of the proposed method is checked against state-of-the-art (e.g., QAM, BPSK) and modulation-free schemes. The parameters considered in the analysis are L.S., MMSE, BER, and PSNR. The details of the comparison are presented in the following sections. Figure 14 and Figure 15 and Table 12, Table 13, Table 14 and Table 15 shows the applicability of channel estimators in OFDM. Looking at OFDM, the BER behavior of LSDFT is quite different, and its value tends to zero as SNR increases. There are two reasons for this. (i) The LS-DFT is used as a Wiener filter when the residual disturbance remains unchanged. (ii) With selectivity, the number of holes in the frequency response increases. The equalizer coefficient Z.F. Fm = 1/H^m is used to estimate the error since the high-frequency equalization stage causes more symbol detection errors.

In Figure 14, the performance of the proposed method is presented taking various modulation schemes into consideration with the inclusion of the PSNR measurement. Under this condition, the PSNR value is observed as the proposed ICEA-DA approach with LS DFT, and provides a PSNR of 42, while other schemes, such as QAM and BPSK, provide modulation values of 9 and 23, respectively. In the case without modulation, the PSNR value is measured as 28.

Figure 15 compares the performance of the DWT-LS-WIENER algorithm with other traditional techniques such as the QAM and BPSK schemes. For these methods, the BER ranges from 103 to 104 with SNR gains of 1 to 11 dB. In the proposed method, BER exists between 101 and 102 with SNR of 10 and 11 dB with significant performance. Hence, the introduced DFT-LS-WIENER approach outperforms compared to conventional approaches.

## 5. Conclusions

The proposed DFT-LS-WIENER estimate performs better than L.S., LS-DFT, MMSE, and MMSE-DFT channel estimation algorithms in terms of BER performance for the channel estimation of MIMO-OFDM. When the number of subcarriers increases, the bit error rate is reduced. As the number of channel taps increases, the bit error rate also increases. As the cyclic prefix length increases, the bit error rate is reduced considerably. When the pilot frequency is increased, the bit error rate is also increased. Including the DFT technique in L.S. and MMSE improves the bit error rate performance. The computational complexing and time complexity of the DFT-LS-WIENER algorithm depends on the number of transmitting and receiving antennas in the system. If the number of transmitting antennae is N and the number of receiving antennae is M, then its time and computational is N*M. The accuracy of the proposed technique is high compared to the L.S., MMSE, LS-DFT, and MMSE-DFT channel estimation algorithms. Compared to the proposed approach, the efficiency of the introduced method attained 85%, 79.5%, 84.9%, and 84.1% compared with L.S., MMSE, LS-DFT, and MMSE-DFT. The proposed DFT-LS-WIENER algorithm is less complex and very easy to implement. The demerits include introducing the Wiener model in channel estimation, increasing the cost of implementation, and extra circuitry required for implementing the Wiener filter model. Our future works include implementing other adaptive filters such as the Kalman and Kernal filters in the channel estimation algorithms. Further, there will be minimal hardware implementation using the proposed methodology, and it will be simultaneously carried out without compromising the proposed methodology’s performance.

## Figures and Tables

**Figure 1 entropy-24-01601-f001:**
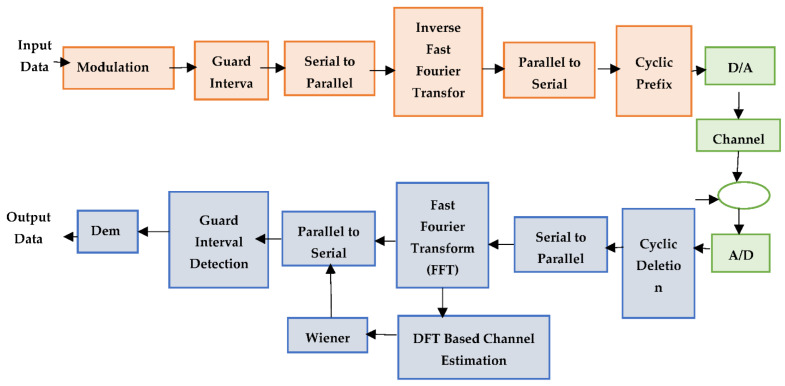
Block diagram of channel estimation in MIMO-OFDM system with Wiener filter model.

**Figure 2 entropy-24-01601-f002:**
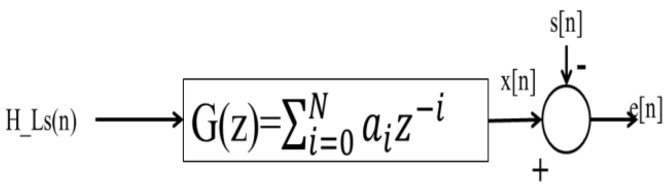
Block diagram view of the F.I.R. Wiener filter with DFT_LS estimates as input.

**Figure 3 entropy-24-01601-f003:**
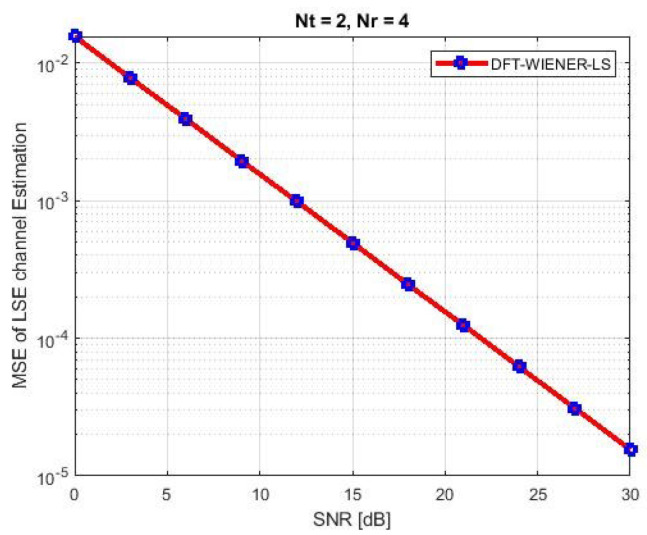
MSE and BER for DFT-LS-WIENER.

**Figure 4 entropy-24-01601-f004:**
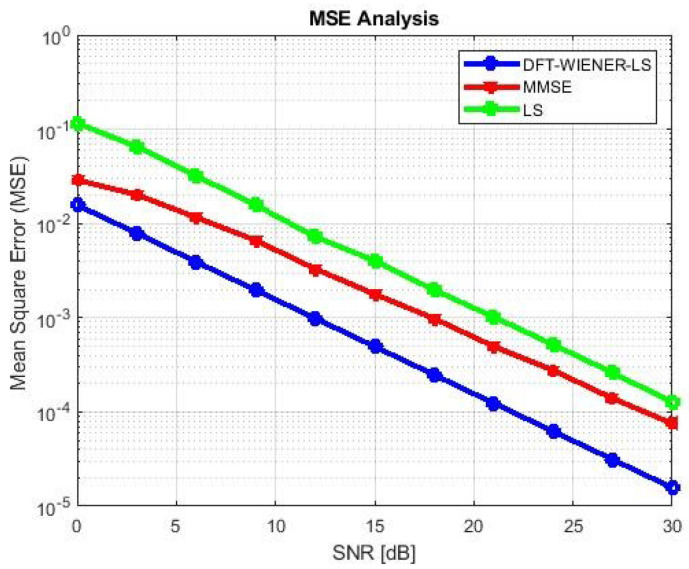
M.S.E. values of L.S., MMSE, and LS-DFT method.

**Figure 5 entropy-24-01601-f005:**
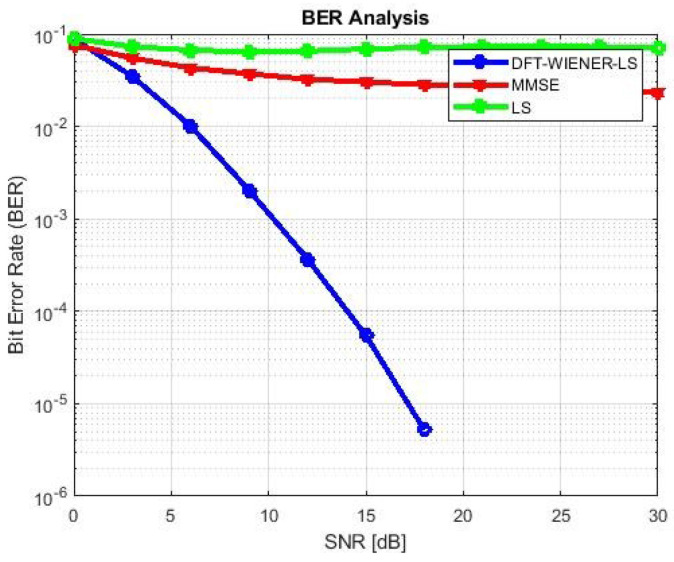
BER values of L.S., MMSE, and LS-DFT method.

**Figure 6 entropy-24-01601-f006:**
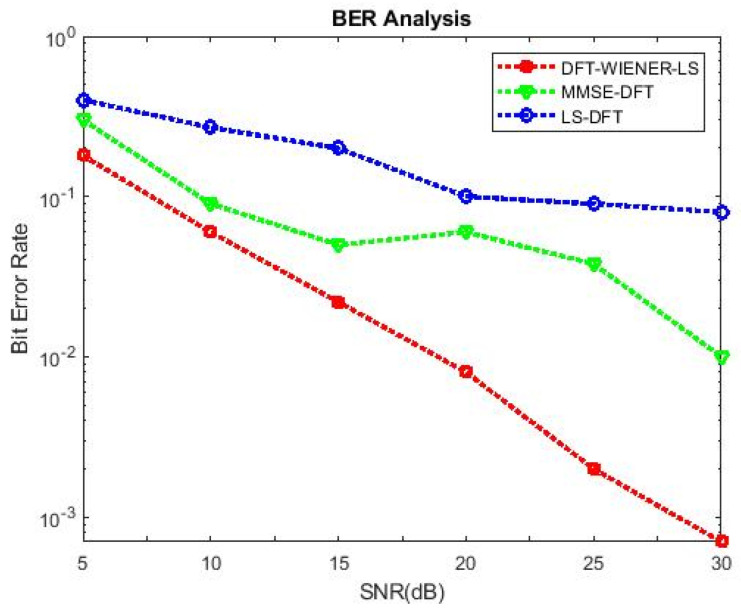
SNR vs. BER for DFT-LS-WIENER with 64 subcarriers.

**Figure 7 entropy-24-01601-f007:**
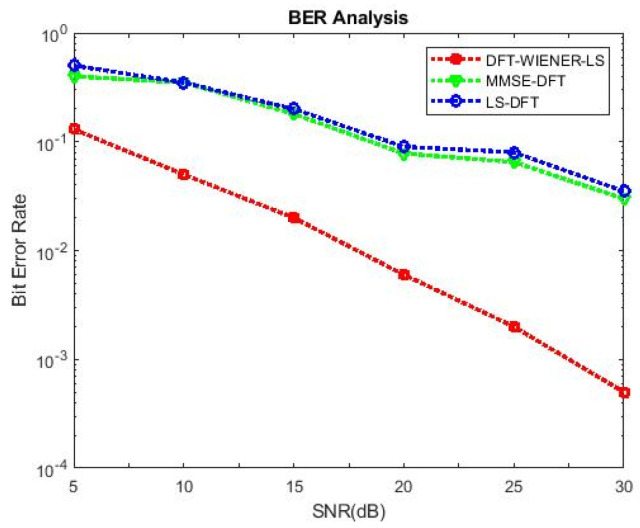
SNR vs. BER for DFT-LS-WIENER with 256 subcarriers.

**Figure 8 entropy-24-01601-f008:**
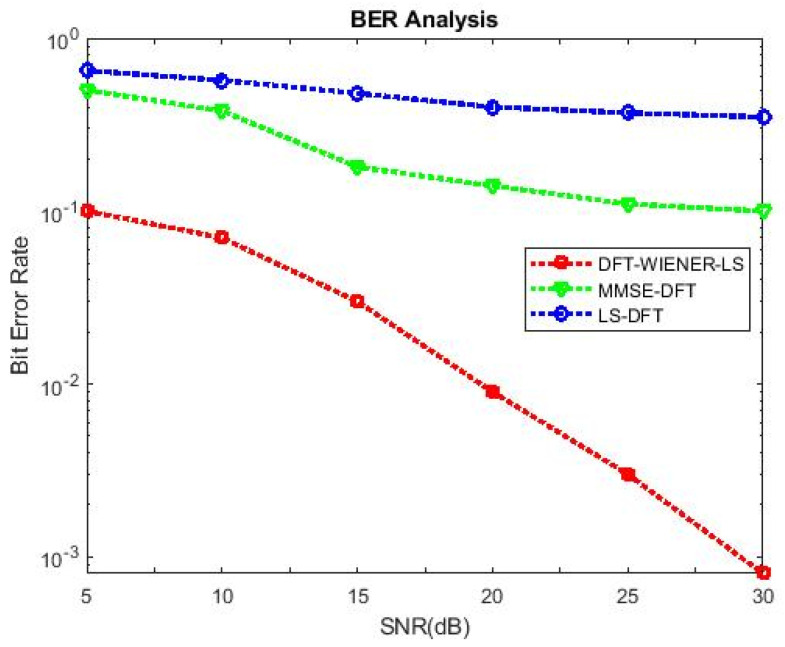
SNR vs. BER for DFT-LS-WIENER with channel taps = 6.

**Figure 9 entropy-24-01601-f009:**
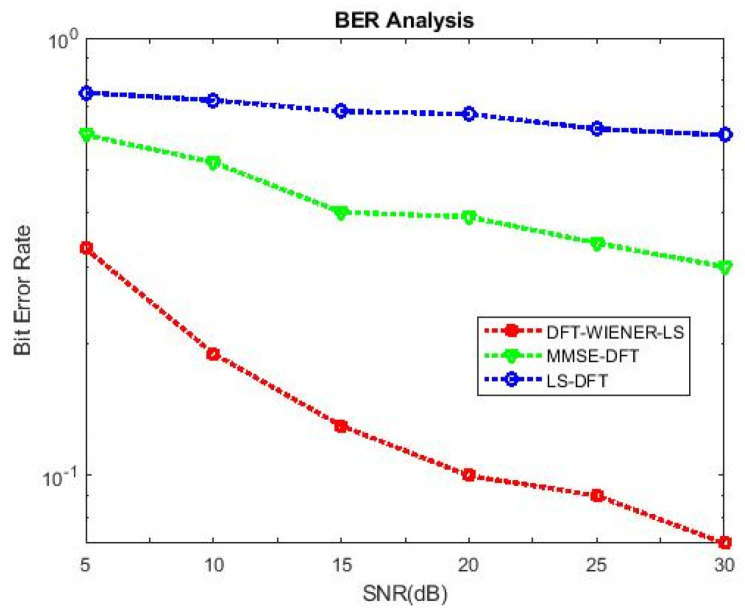
SNR vs. BER for DFT-LS-WIENER with channel taps = 10.

**Figure 10 entropy-24-01601-f010:**
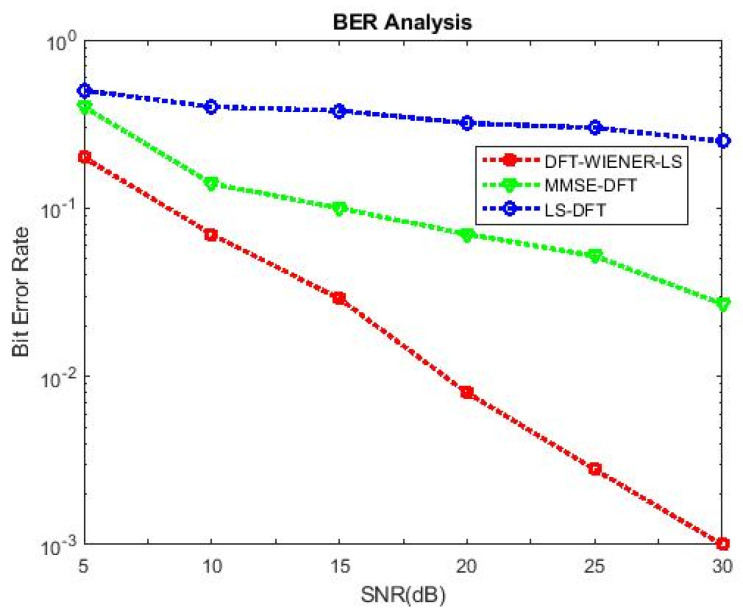
SNR vs. BER for DFT-LS-WIENER with cyclic prefix length = 1/8th of K.

**Figure 11 entropy-24-01601-f011:**
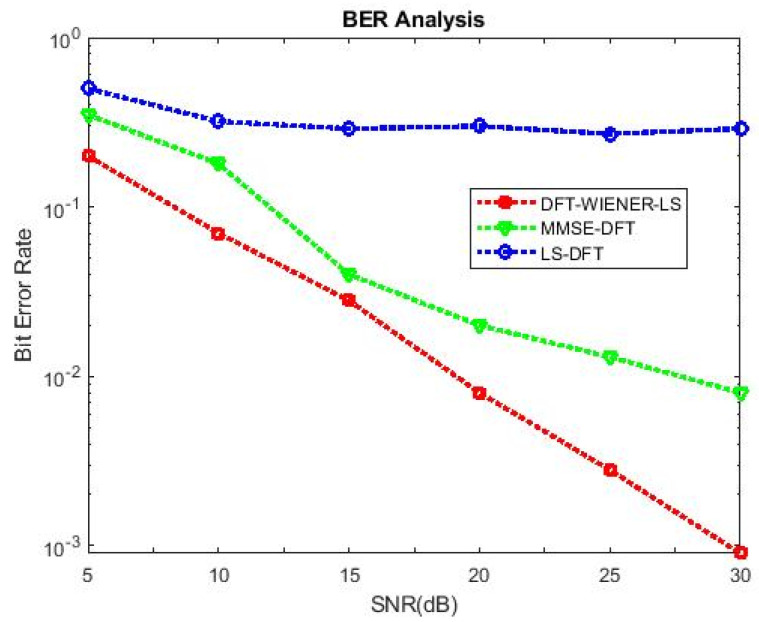
SNR vs. BER for DFT-LS-WIENER with cyclic prefix length = 1/2th of K.

**Figure 12 entropy-24-01601-f012:**
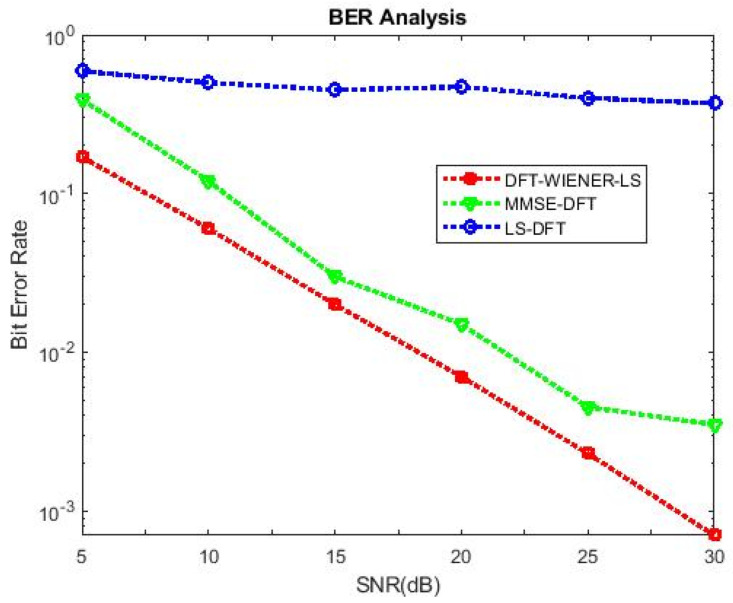
SNR vs. BER for DFT-LS-WIENER with pilot frequency = 16.

**Figure 13 entropy-24-01601-f013:**
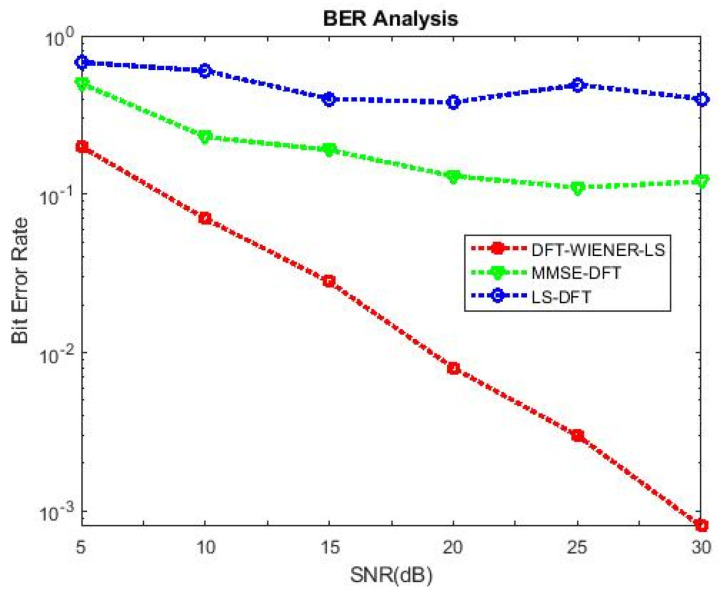
SNR vs. BER for DFT-LS-WIENER with pilot frequency = 32.

**Figure 14 entropy-24-01601-f014:**
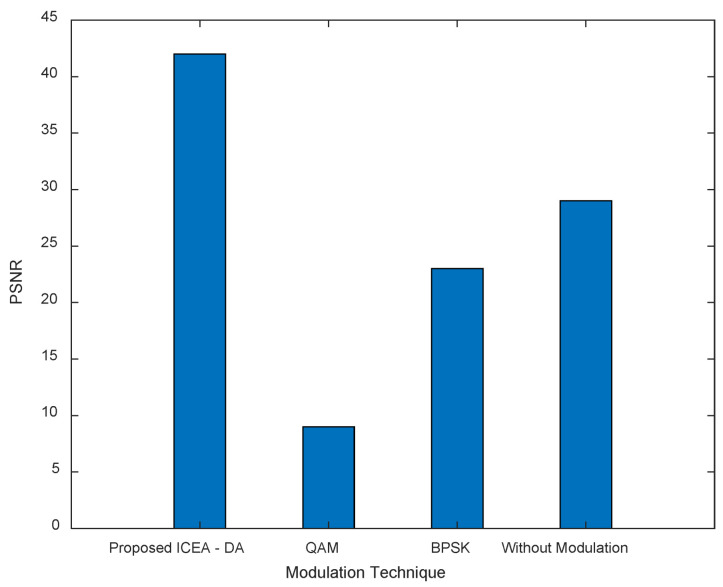
Comparison of PSNR for proposed with other techniques.

**Figure 15 entropy-24-01601-f015:**
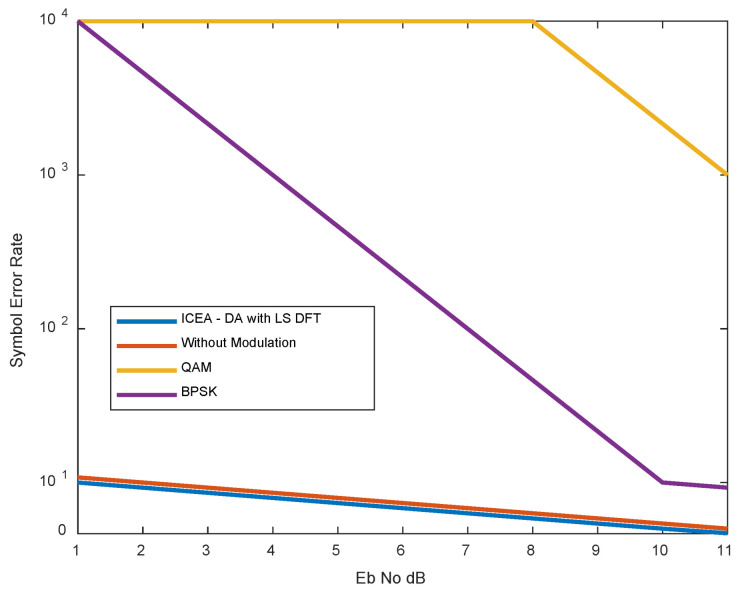
Comparison of S.E.R. for proposed with existing technique.

**Table 1 entropy-24-01601-t001:** Simulation parameters.

No FFT Points	1085
Length of Cyclic Prefix	648
Total no of Subcarriers	5385
Total no of Symbol	289
Pilot Arrangement	Recursive type
Pilot Constellation	QPSK
Data Constellation	QPSK
Bandwidth	9 MHZ
SNR Range	0–30 dB
Data subcarrier	64, 128, 256, 512
Pilot spacing	16, 32
Number of Iterations	300
Denoising model	Wiener filter

**Table 2 entropy-24-01601-t002:** Analysis of optimal power allocation methods.

S. No	Authors Name	Methods	Merits	Demerits
1.	Lee, J et al. [7]	Hybrid mmWave MIMO-OFDM Channel Estimation	It yields substantial system throughput gains with reduced computational overhead	Necessary to reduce SNRs when huge numbers of the antenna are used
2.	Qiao, G et al. [8]	Bit and power allocation algorithm	Reduced Bit Error Rate	Still, it faces problems with low complexity bit and a power allotment
3.	Berraki, D.E et al. [10]	New Adaptive greedy algorithm strategy	Reduced M.S.E.	Superior techniques are necessary to boost power allocation with reduced complexity
4.	Khan, et al. [41]	Spatial partitioning with coalitional game theory (SPCGT)	Improved power allocation	The convergence rate has to be improved
5.	Dhanasekaran et al. [31]	Energy-efficient equal power allocation method using G.A.	Optimal performance with minimal complexity	M.S.E. is a concern
6.	Kumar, P.S. et al. [42]	Channel Prediction based Temporal Multiple Sparse Bayesian Learning for Channel Estimation	Considerably increased the throughput	It faces problems with local optimum
7.	Fereydouni, A.R et al. [49]	Singular value decomposition	Improved diversity gains	Advanced techniques are necessary to yield optimal power allocation

**Table 3 entropy-24-01601-t003:** MSE and BER of hybrid DFT-LS-WIENER.

SNR (dB)	BER	MSE
0	0.088385	0.015678
3	0.034779	0.007881
6	0.010104	0.003916
9	0.002036	0.001961
12	0.000367	0.000987
15	5.47 × 10^−5^	0.000494

**Table 4 entropy-24-01601-t004:** M.S.E. of LS AND MMSE.

SNR (dB)	M.S.E. of MMSE	MSE of L.S.
0	0.028968	0.115917
3	0.020203	0.065559
6	0.011584	0.031985
9	0.006628	0.015689
12	0.003304	0.007218
15	0.001777	0.004017

**Table 5 entropy-24-01601-t005:** BER of LS AND MMSE.

SNR (dB)	BER of MMSE	BER of LS
0	0.075906	0.089109
3	0.054953	0.073078
6	0.043031	0.066734
9	0.037188	0.064219
12	0.032375	0.065094
15	0.030234	0.069031

**Table 6 entropy-24-01601-t006:** M.S.E. and BER values of DFT, Harmonic retrieval, and LS-DFT method.

Parameters	Harmonic Retrieval Method	LS-DFT	DFT-LS-WEINER
MSE	0.1689	0.004017	0.4310
BER	0.08654	0.069031	0.4198

**Table 7 entropy-24-01601-t007:** Conventional complexity of DFT, Harmonic retrieval, and LS-DFT method.

Number of Blocks	Complex Computation	Harmonic Retrieval Method	LS-DFT	DFT-LS-WEINER
4	Rotation factors	427,930 (23.3%)	855,900 (46.6%)	19.3%
	Multiplication	126,362 (52.9%)	138,650 (48.2%)	39.6%
	Addition	1,104,500 (28.1%)	1,850,400 (47.1%)	21.4%
8	Rotation factors	102,040 (20.4%)	267,893 (40.1%)	16.5%
	Multiplication	299,008 (46.1%)	303,104 (51.4%)	40%
	Addition	237,568 (20.2%)	245,760 (45.5%)	17.1%

**Table 8 entropy-24-01601-t008:** SNR vs. BER performance with the Number of subcarriers varied.

Bit Error Rate (BER)						
SNR (dB)	No. of Subcarrier	LS	LS-DFT	MMSE	MMSE-DFT	DFT-LS-WEINER
15	64	0.533	0.5	0.4167	0.35	0.02285
15	128	0.4083	0.25	0.244	0.2333	0.01871
15	256	0.3417	0.2375	0.2313	0.1908	0.01678
15	512	0.3001	0.20	0.221	0.1591	0.01601

**Table 9 entropy-24-01601-t009:** SNR vs. BER performance with the Number of channel taps varied.

Bit Error Rate (BER)						
SNR (dB)	No. of Subcarrier	LS	LS-DFT	MMSE	MMSE-DFT	DFT-LS-WEINER
15	2	0.4083	0.25	0.333	0.2333	0.01871
15	6	0.591	0.399	0.201	0.190	0.083
15	10	0.7167	0.675	0.4417	0.4	0.1192

**Table 10 entropy-24-01601-t010:** SNR vs. BER performance with the length of cyclic prefix varied.

Bit Error Rate (BER)						
SNR (dB)	No. of Subcarrier	LS	LS-DFT	MMSE	MMSE-DFT	DFT-LS-WEINER
15	1/8th of K	0.5667	0.4417	0.2333	0.21	0.02583
15	1/4th of K	0.525	0.175	0.175	0.1583	0.02669
15	1/2th of K	0.35	0.05833	0.05833	0.03	0.02683

**Table 11 entropy-24-01601-t011:** SNR vs. BER performance with pilot frequency varied.

Bit Error Rate (BER)						
SNR (dB)	No. of Subcarrier	LS	LS-DFT	MMSE	MMSE-DFT	DFT-LS-WEINER
15	16	0.5083	0.4083	0.175	0.1667	0.02641
15	32	0.5240	0.5240	0.291	0.199	0.099

**Table 12 entropy-24-01601-t012:** Efficiency of DFT-LS WIENER over L.S., LS-DFT, MMSE, MMSE-DFT when the Number of subcarriers varied.

Efficiency of DFT-LS WIENER				
No. of Subcarrier	L.S. (%)	LS-DFT (%)	MMSE (%)	MMSE-DFT (%)
64	95.7	95.4	94.5	93.4
128	95.4	92.4	92.33	91.9
256	96.6	92.99	92.78	91.2
512	94.6	92.92	92.70	89.9

**Table 13 entropy-24-01601-t013:** Efficiency of DFT-LS WIENER over L.S., LS-DFT, MMSE, MMSE-DFT when Number of channel taps varied.

Efficiency of DFT-LS WIENER				
Channel Taps	LS (%)	LS-DFT (%)	MMSE (%)	MMSE-DFT (%)
2	95.41	95.5	92.3	91.9
6	85.9	79.1	58.7	56.3
10	83.36	82.3	73.01	70.2

**Table 14 entropy-24-01601-t014:** Efficiency of DFT-LS WIENER over L.S., LS-DFT, MMSE, MMSE-DFT when length of cyclic prefix varied.

Efficiency of DFT-LS WIENER				
Cyclic Prefix Length	L.S. (%)	LS-DFT (%)	MMSE (%)	MMSE-DFT (%)
1/8th of K	95.4	94.1	88.9	87.7
1/4th of K	94.91	93.7	84.7	83.1
1/2th of K	92.3	90.2	54	40

**Table 15 entropy-24-01601-t015:** Efficiency of DFT-LS WIENER over L.S., LS-DFT, MMSE, MMSE-DFT when pilot frequency varied.

Efficiency of DFT-LS WIENER				
Pilot Frequency	L.S. (%)	LS-DFT (%)	MMSE (%)	MMSE-DFT (%)
16	85	79.5	84.9	84.1
32	85.6	81.1	65.9	50.2

## Data Availability

The study did not report any data.

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
