# Peer review of "Discrete Fourier Transform with Denoise Model Based Least Square Wiener Channel Estimator for Channel Estimation in MIMO-OFDM"

_entropy, 2022, doi:10.3390/e24111601_

Round 1

Reviewer 1 Report

The key contributions claimed by the authors of this manuscript:

“The proposed DFT-LS-WIENER estimate performs better than LS, LS-DFT, MMSE, and MMSE-DFT channel estimation algorithms in terms of BER performance for channel estimation of MIMO-OFDM. The accuracy of the proposed technique is high compared to LS, MMSE, LS-DFT, and MMSE- DFT channel estimation algorithms. The proposed DFT-LS-WIENER algorithm is less complex and very easy to implement”

The English writing is good but there is a lack of presentation quality of this article. The authors have properly discussed the system model, and problem statement. The results section is also provided with sufficient discussion. There are some major concerns that authors must address:

1.     In the Abstract, authors must provide an exact numerical comparison of their proposed approach with others.

2.     In Abstract, what is LS, MMSE, DFT, QPSK? It is suggested to check each abbreviation and define at the first place of appearance.

3.     Abstract must be revised and writing quality must be strengthened.

4.     Line 38; what is CSI? Channel State Information then you must define it to make it easier for readers.

5.     The organizational quality of this study is not good.

6.     Authors should provide organization of this study in the end of introductions section.

7.     What is the purpose of Literature Survey in a research article? It does not fit. It is suggested to summarize the provided discussion in a Table as readers can fastly understand the research findings from Tables.

8.     The figures quality of this article is really bad.

9.     System Model and Problem Formulation are fine. Must provide a good diagram for the system model.

10.                        How to justify your contribution with blur results?

11.                        The authors should carefully check equations and define all included variables.

12.                        For results, it is suggested to provide more comprehensive discussions.

13.                        In the Table, proper units must be added. For instance, in Tables 3-4, what’s the unit for SNR?

14.                        In reference, several references are outdated. It is suggested to add maximum reference literature from the latest research contributions from 2020-2022.

15.                        All the listed results are blur. Unable to justify it. The authors must provide supplementary data to justify it.

Reviewer 2 Report

Please consider the following remarks:

1)  Main research findings could be verified at abstract section with numerical results.

2) Please put references in numerical order in text.

3) Please explain all acronyms appeared in text (e.g. NAMP).

4) Please improve the quality of Figure 1.

5) After equation 8 the authors go to equation 13. Please correct.

6) Line 310:  Please remove the word "procedure". It can be included at section 3.5 heading.

7) Line 359: Please include subheading 4.1.1.

8) Line 363: Equation number?

9) Line 366: Please include subheading 4.1.2.

10) Tables 7, 8 are not explained.

11) Figures 7, 8, 9, 11, 12, 13 are not explained.

12) Please rewrite conclusion section focusing on main findings and future work. 

Round 2

Reviewer 1 Report

Dear authors,

Thank you very much for your efforts. You have addressed my concerns very well. However, I suggest improving the quality of the results shown in Figures 6-13. These results must be provided in high-resolution quality. For instance, compare the quality of Figure 13 with  Figure 14 and you can easily find the difference. 

All the best
